# Research of multi-label text classification based on label attention and correlation networks

**Ling Yuan**[1], **Xinyi Xu**[1], **Ping Sun**[2]*, **Hai ping Yu**[2], **Yin Zhen Wei**[2], **Jun jie Zhou**[2]

**1** School of Computing Science and Technology, Huazhong University of Science and Technology, Wuhan, Hubei, China, **2** Wuhan Vocational College of Software and Engineering (Wuhan Open University), Wuhan, Hubei, China

\* ppsun@126.com

**Data Availability Statement:** All the datasets utilized in this paper are open source, which are available at: AAPD: https://github.com/lancopku/SGM RCV1-v2: https://trec.nist.gov/data/reuters/

## Abstract

Multi-Label Text Classification (MLTC) is a crucial task in natural language processing. Compared to single-label text classification, MLTC is more challenging due to its vast collection of labels which include extracting local semantic information, learning label correlations, and solving label data imbalance problems. This paper proposes a model of Label Attention and Correlation Networks (LACN) to address the challenges of classifying multi-label text and enhance classification performance. The proposed model employs the label attention mechanism for a more discriminative text representation and uses the correlation network based on label distribution to enhance the classification results. Also, a weight factor based on the number of samples and a modulation function based on prediction probability are combined to alleviate the label data imbalance effectively. Extensive experiments are conducted on the widely-used conventional datasets AAPD and RCV1-v2, and extreme datasets EUR-LEX and AmazonCat-13K. The results indicate that the proposed model can be used to deal with extreme multi-label data and achieve optimal or suboptimal results versus state-of-the-art methods. For the AAPD dataset, compared with the suboptimal method, it outperforms the second-best method by 2.05% ∼ 5.07% in precision@k and by 2.10% ∼ 3.24% in NDCG@k for k = 1, 3, 5. The superior outcomes demonstrate the effectiveness of LACN and its competitiveness in dealing with MLTC tasks.

## 1 Introduction

Recently, as the volume of data continues to grow exponentially, it has become crucial to extract meaningful information from vast amounts of data. Text classification technology has emerged as a solution to this challenge, and it is mainly categorized into single-label and multi-label text classifications. Among them, Multi-Label Text Classification (MLTC) is more aligned with real-world requirements as it offers finer granularity and multiple label levels. This enables a comprehensive description of a document from various perspectives. As a vital natural language processing task, MLTC is widely used in topic recognition [1], question-

reuters.html EUR-LEX: http://manikvarma.org/downloads/XC/XMLRepository.html.

**Funding:** This paper is funded by the National Natural Science Foundation of China under Grant No.62272180, Hubei Provincial Teaching and Research Project for Higher Education Institutions (No.2022570), Wuhan Education Science Planning Project (No.2022C151), Wuhan Vocational College of Software and Engineering Research Startup funding project (Grant No.KYQDJF2023004), Wuhan Vocational College of Software and Engineering 2023 Doctor Team Science and Technology Innovation Platform Project (No. BSPT2023001).

**Competing interests:** The authors have declared that no competing interests exist.

answer systems [2], sentiment analysis [3], text classification [4], search [5] and text summarization [6]. However, MLTC poses challenges with larger text volumes and extensive label sets in the big data era. Therefore, it has become essential to develop effective multi-label classifiers for various applications. To solve the MLTC problem, we investigate a large number of previous studies and summarize them from three primary aspects as follows:

Firstly, we analyze the perspective of MLTC based on known documents and labels. Currently, existing MLTC algorithms utilize traditional machine learning and deep learning. Traditional machine learning methods such as BR [7] and CC [8] simplify MLTC to single-label tasks but encounter difficulties with large label spaces. Deep learning methods, on the other hand, suffer from continuous data and gradient vanishing problems. However, Bi-LSTM [9] and GRU [10] overcome these challenges with gating mechanisms. Recent models have combined CNN and RNN and added attention mechanisms [11] for better text feature extraction. A recent study proposed the hybrid CNN-LSTM [12] model for feature detection and spatial generalization of CNN, which increases efficiency.

Secondly, there are identical subsets between labels, and some researchers have worked to find the correlation between labels accurately. The current algorithms used to determine label correlation in the pair relation can be classified into four categories: one-to-many, tree, sequence generation, and label embedding. One-to-many methods are effective for small label spaces, such as BR [7] and CC [8]. Tree methods, like Attention-XML [13] and Hierarchy-Aware Global [14], can handle complex label relationships but result in error accumulation. Sequence generation methods, such as Seq2Seq [15], capture label correlations but are sensitive to order. Seq2Set [16] addresses this issue with reinforcement learning. Label embedding methods, such as LEAM [17], utilize attention to establish compatibility between text and labels, resulting in joint embeddings.

Thirdly, the imbalance problem of label data is also an excellent research direction. There are several approaches including resampling, classifier adaptation, ensemble, and cost-sensitive methods. Resampling involves both under-sampling, which removes head label samples, and over-sampling, which increases tail label samples. Examples of this method include LP-RUS [18], LP-ROS [18] based on LP [19], and ML-SMOTE [20]. Besides, BBN [21] uses a bilateral branch network for classifier learning. Classifier adaptive methods directly use imbalanced data to train models, and then use machine learning methods such as basing on a min-max modular support vector machine network [22] or increasing the complexity of neural networks to adapt to imbalanced distributions [23]. Ensemble methods integrate multiple models for optimal prediction [24]. Cost-sensitive methods assign varied costs to labels, like SOSHF [25] and DB loss [26], balancing labels through clustering and reweighting.

With the gradual deepening of research, MLTC encounters several complex challenges. Existing MLTC methods still have problems in extracting local semantic information, learning label correlation and solving label data imbalance. Firstly, current methods overlook label text information in local semantic extraction. Deep neural networks such as CNN and RNN have been able to obtain complex semantic representations from documents and perform well in single-label text classification However, with large label spaces in MLTC for longer documents, these methods that only consider the content of documents fail to show the differences in the focus between different labels. Secondly, labels in MLTC tasks are often correlated with each other, and the huge amount of label space makes it difficult to mine label associations. Early MLTC methods ignore label correlation or limit to small label sets. To capture label correlation, existing algorithms such as classifier chain and Seq2Seq [15] sequence model rely heavily on the input label order, and there are problems of overfitting and error accumulation. Thirdly, label imbalance is a pressing issue. Although a large amount of research is devoted to solving this problem, due to co-occurrence and huge label space in MLTC, traditional

**Table 1. Comparison of the reviewed papers.**

| Source | Research method | Text representation | Label prediction |
|---|---|---|---|
| LSAN [11] | Label-specific attention network | Employ the self-attention mechanism to extract label-specific document representations and use an adaptive fusion strategy to produce a comprehensive document representation. | Build the multi-label text classifier via a multilayer perceptron with two fully connected layers. |
| CNN-LSTM [12] | A hybrid CNN-LSTM model | Use the TF-IDF weighting method and make word vectors as input data for text representation. Also, employ the CNN model's convolutional and max pooling layers for high-level feature extraction. | Use LSTM to enhance CNN to improve the accuracy and the competitive search algorithm (CSA) to improve the LSTM hyperparameters. |
| Attention-XML [13] | Label tree-based deep learning model | Use the multi-label attention mechanism for label-specific document representations. | Use a shallow and wide top-down hierarchical training PLT model to divide labels for classifier and beam search algorithm to improve the prediction efficiency. |
| LEAM [17] | The method that jointly learns information of words and sentiment labels | Use text-label compatibility for text representations by jointly embedding words and labels in the same latent space | Use a novel attention-based method(AW ave.) for better usage of the categorical lexicons |
| BBN [21] | A bilateral branch network for representation learning and classifier learning simultaneously | Use a unified Bilateral-Branch Network (BBN) model to take care of both representation learning and classifier learning for exhaustively boosting long-tailed recognition | Use a novel cumulative learning strategy for adjusting the bilateral learning and couplingwith the BBN model training. |
| DB loss [26] | Distribution-Balanced Loss for the multi-label recognition problems | Use re-sampling and re-weighting to better learn labeling specific document representations. | Use a new way to re-balance the weights that takes into account the impact of label co-occurrence and a negative tolerant regularization to mitigate the over-suppression of negative labels. |

resampling and reweighting techniques are ineffective and even reduce the robustness of the model. The following Table 1 compares the methods proposed by some typical literature in recent years.

To address these problems, this paper proposes a model based on Label Attention and Correlation Networks (LACN) to effectively address the problems of label correlations and data imbalance in MLTC. Firstly, LACN captures the important local features in the document that are relevant to the label using the document attention mechanism. At the same time, the label attention mechanism is utilized to calculate the semantic connection between the document words and the label. After obtaining these two text representations, they are combined via an adaptive fusion mechanism to create a label-specific text representation that enables more accurate classification. Subsequently, the text representation based on specific labels is mapped into an initial vector to represent its classification result using a fully connected layer and an output layer. Then, the initial classification result is enhanced through label distribution-based correlation networks which are implemented by stacking multiple correlation residual blocks. The above steps can alleviate the problem of overfitting head labels caused by label imbalance to a certain extent, but the issue of tail labels being suppressed by head labels persists. To address this problem, we propose a label imbalance loss function that assigns different weights to samples of different labels to make the model pay attention to minority label samples. Additionally, the modulation function is studied based on the difficulty of label classification and selective suppression of negative samples.

In summary, the main contributions of this paper are as follows:

1. While feature extraction is performed on the original document, the attention mechanism is used to identify the most relevant text information for a particular label. Thus, the most relevant classification discriminative information is obtained for different labels.

2. This paper proposes a correlation residual network that is based on label distribution. In this approach, input text sequences are labeled with the most appropriate subset of labels

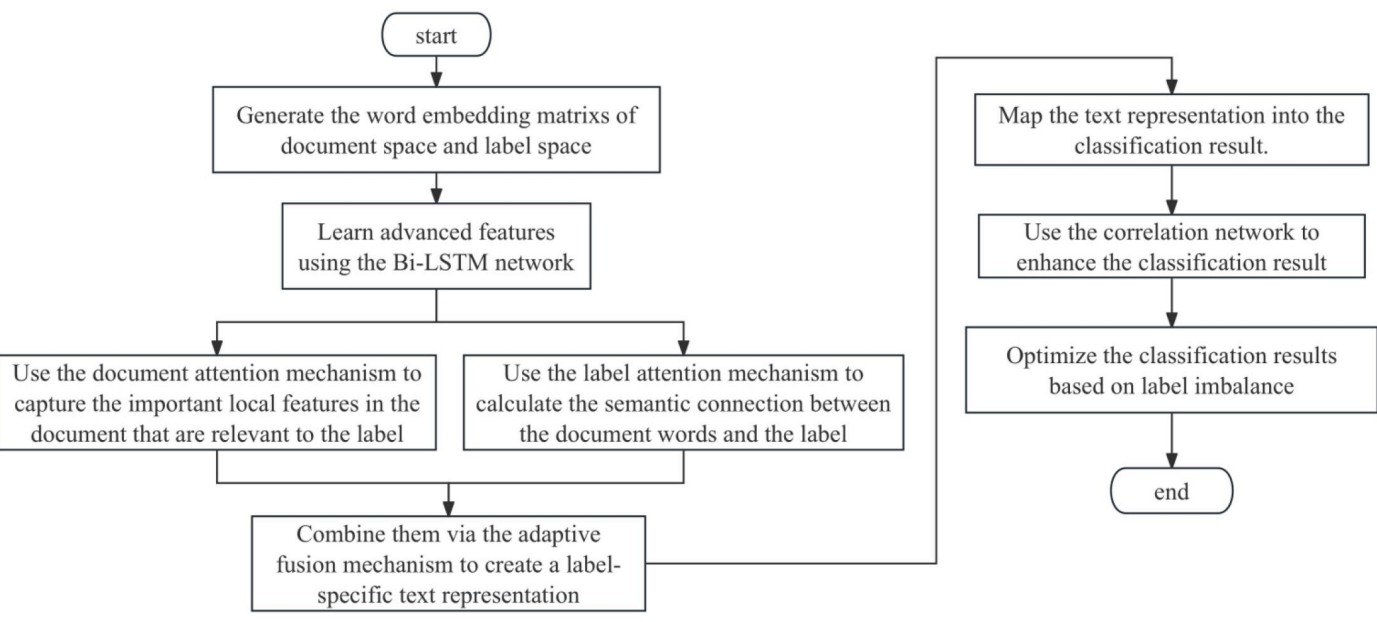

**Fig 1. Flowchart of LACN.**

from the label set. The idea is to leverage relevant knowledge to learn label correlation and thus reduce the probability of label misclassification.

3. To address the problem of data and sample imbalance, this paper proposes a weighting factor based on the number of labels and a modulation function based on the classification result to change the cost-sensitive loss function.

The rest of the paper is organized as follows. Section 2 details the framework of the model, including text representation, correlation networks, and label prediction optimization. Section 3 evaluates the model on a benchmark dataset, comparing it with existing models and conducting an ablation experiment of crucial components. Section 4 summarizes the paper, discusses limitations, and outlines future research directions.

## 2 Methods

This section discusses the proposed LACN method, as shown in the flowchart of Fig 1, which mainly involves three parts: text representation based on document and label attention, correlation networks based on label distribution, and optimization of classification results based on label imbalance. The overall framework of LACN is illustrated in Fig 2.

### 2.1 Text representation based on document and label attention

The document and label information-based text representation network model mainly includes document and label word embeddings, Bi-LSTM for long-distance feature extraction, document attention mechanism, label attention mechanism, and adaptive text representation fusion mechanism.

**2.1.1 Word embedding of document and label.** In a document $S = \{x_1, x_2, \ldots, x_T\}$ with $T$ words and a label set $Y = \{y_1, y_2, \ldots, y_l\}$ with $l$ labels, each word or label is initially represented as a binary vector $v^i = \{0, 0, 0, \ldots, 1, \ldots, 0\}$. Word2vec [27] is used to map sparse vectors $v^i$ to

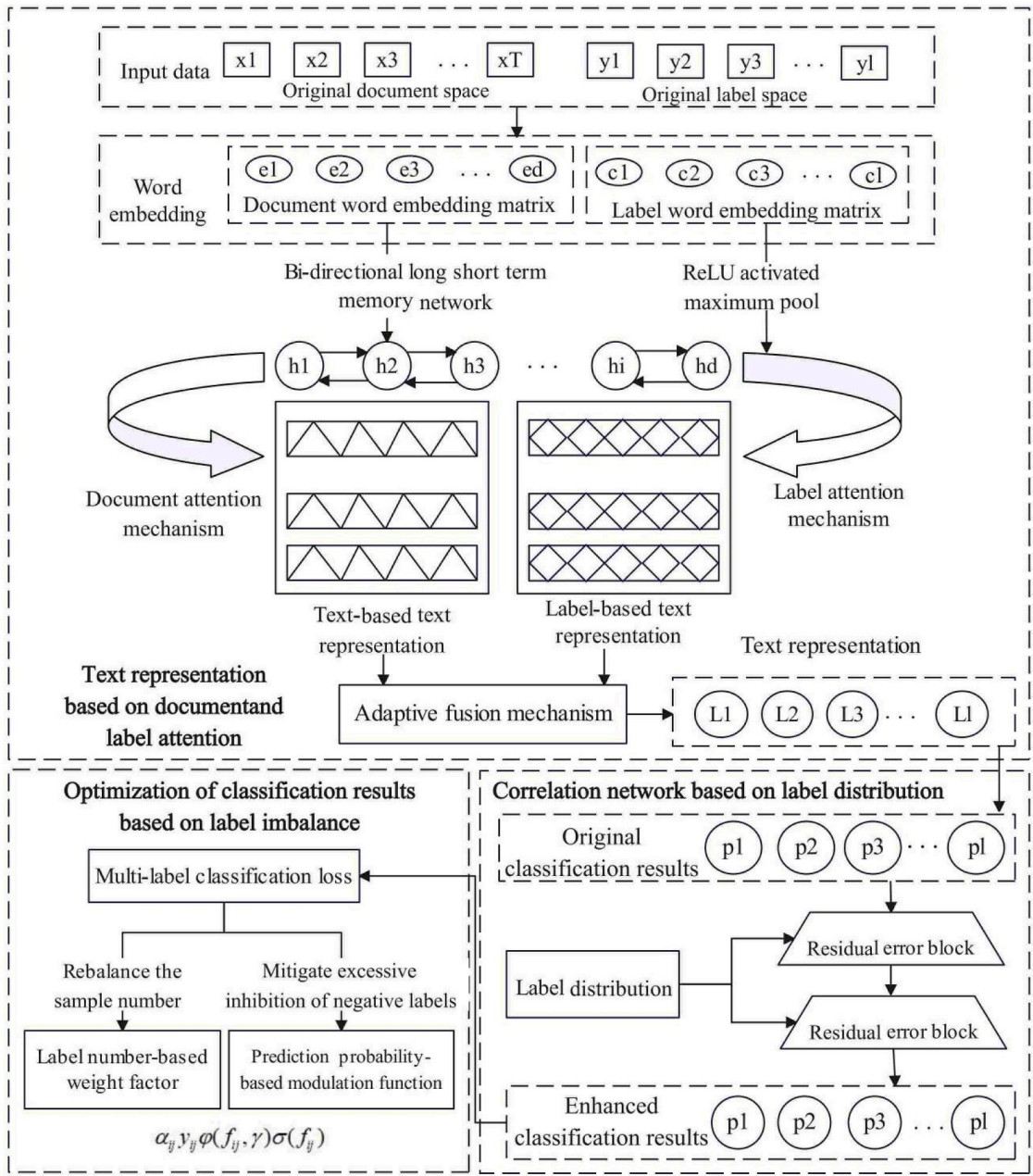

**Fig 2. Overall framework of LACN.**

dense vectors in a low-dimensional space for facilitating word similarity analysis. The generated neural network weight $W^{wrd} \in R^{d^w|V|}$ convert $v^i$ into word embedding vectors:

$$e_i = W^{wrd}v^i \tag{1}$$

where $W^{wrd}$ is the learned parameter. $V$ donates size-fixed vocabulary and $d^w$ represents word embedding size. Through training, the document word embedding matrix $emb_s = \{e_1, e_2, \ldots, e_T\}$ is obtained from the input document. For label word embedding matrix $label\_emb_l = \{c_1, c_2, \ldots, c_l\}$, the label embedding representation can directly use word vector in the text if the

label appears in the dataset. Otherwise, the samples related to the label are randomly selected, and the word vectors in the samples are used as the initial embedding representation.

**2.1.2 Long-distance feature extraction based on Bi-LSTM network.** In this paper, Bi-LSTM [9] is used to extract important information from input documents and convert input word embedding matrix $emb_s = \{e_1, e_2, \ldots, e_T\}$ to feature matrix $H$. For the input word embedding vector $e_t$ at time $t$, use LSTM with gate signal to self-update and get hidden state $h_t$. The gate signal $f_t$ of the forgetting gate is trained by weight $W_f$ to selectively forget the internal state $c_{t-1}$ at the previous time. The gate signal $i_t$ of the input gate is trained by weight $W_i$ to selectively memorize the memoryless internal state $C_t$ at the current time. The gating signal $o_t$ of the output gate is trained by weight $W_o$ to decide the feature value $h_t$ of the current state $c_t$ output. Where $\bullet$ represents matrix multiplication, $\oplus$ represents matrix addition, $\otimes$ represents Hadamard product, and $\sigma$ represents sigmoid activation function.

$$f_t = \sigma(W_f \bullet [h_{t-1}, e_t] + b_f) \tag{2}$$

$$i_t = \sigma(W_i \bullet [h_{t-1}, e_t] + b_i) \tag{3}$$

$$o_t = \sigma(W_o \bullet [h_{t-1}, e_t] + b_o) \tag{4}$$

$$C_t = tanh(W_c \bullet [h_{t-1}, e_t] + b_c) \quad c_t = f_t \otimes c_{t-1} \oplus i_t \otimes C_t \tag{5}$$

$$h_t = o_t \otimes tanh c_t \tag{6}$$

Word embedding vector feature $h_t$ combines the hidden state $\overrightarrow{h_t}$ of the backward left sequence context and the hidden state $\overleftarrow{h_t}$ of the forward right sequence context:

$$h_t = (\overrightarrow{h_t}, \overleftarrow{h_t}) \quad \overrightarrow{h_t} = LSTM(\overrightarrow{h_{t-1}}, e_t) \quad \overleftarrow{h_i} = LSTM(\overleftarrow{h_{t-1}}, e_i) \tag{7}$$

Concatenate parallel and opposite hidden states to transform input word embeddings into a feature matrix $H$. Represent document $S$ using Bi-LSTM [9] networks:

$$H = (\overrightarrow{H}, \overleftarrow{H}) \quad \overrightarrow{H} = (\overrightarrow{h_1}, \overrightarrow{h_2}, \ldots, \overrightarrow{h_T}) \quad \overleftarrow{H} = (\overleftarrow{h_1}, \overleftarrow{h_2}, \ldots, \overleftarrow{h_T}) \tag{8}$$

**2.1.3 Document attention mechanism.** Feature extraction from a document involves using LSTM hidden state $H = \{h_1, h_2, \ldots, h_T\}$ as input to calculate weight vector $a$:

$$a = softmax(W_{s2} tanh(W_{s1} H^T)) \tag{9}$$

where $W_{s1} \in R^{d_a \times 2u}$ and $W_{s2} \in R^{d_a}$ are training weight matrix and parameter vector, and $d_a$ is the selective hyperparameter. The $H \in R^{2u \times n}$ is used to obtain the attention vector $a \in R^n$. The normalization function softmax is used to ensure that the calculated weights sum is one and further highlight the weight of important information. According to the calculated weight vector $a$, the LSTM hidden state $H = \{h_1, h_2, \ldots, h_T\}$ is weighted and averaged to obtain the vector representation $m$ of the document $S$:

$$m = aH^T \tag{10}$$

Vector representation $m$ emphasizes specific document sections via trained weights and parameters. To capture the overall semantics of a document, multiple vectors focus on various parts, requiring multiple attention jumps for each label. Specifically, the trained parameter

vector $W_{s2} \in R^{d_a}$ is expanded into a parameter matrix $W_{s2} \in R^{l \times d_a}$, and the obtained weight vector $a \in R^n$ is expanded into a weight matrix $A^s \in R^{l \times n}$. The calculation method is as follows:

$$A^s = softmax(W_{s2} tanh(W_{s1}H^T)) \tag{11}$$

The $j$-th column $A_j^s \in R^n$ of the weight vector $A^s$ represents the contribution of the document words to label $j$. The most relevant semantic information of each label in the document to label $j$ can be obtained with $A_j^s$:

$$M_j^s = A_j^s H^T \tag{12}$$

Embedding vector $m \in R^{n \times 2u}$ expands to matrix $M^s \in R^{l \times 2u}$, where $M_j^s \in R^{2u}$ is the $j$-th column of the embedding matrix, representing optimal text representation for label $j$. The transition is from LSTM hidden state matrix $H = \{h_1, h_2, \ldots, h_T\}$ to document-based representation, using weight matrices $W_{s1} \in R^{d_a \times 2u}$ and $W_{s2} \in R^{d_a \times l}$ to derive weight vector $A^s \in R^{l \times n}$ for each label and achieving final representation $M^s \in R^{l \times 2u}$ via weighted average:

$$M^s = A^s H^T \tag{13}$$

**2.1.4 Label attention mechanism.**   First input the label word embedding matrix $label\_emb_l = \{c_1, c_2, \ldots, c_l\}$, where $C \in R^{l \times u}$, $l$ is the label number and $u$ is the dimension of the trained embedding matrix. The compatibility $\overline{A} \in R^{l \times 2n}$ of label-word pairs is measured by dot product:

$$\overline{A} = CH = \{\overrightarrow{A}, \overleftarrow{A}\} = (C\overrightarrow{H}, C\overleftarrow{H}) \tag{14}$$

where $\overrightarrow{A} \in R^{l \times n}$ and $\overleftarrow{A} \in R^{l \times n}$ respectively represent the context semantic relationship between words and label texts, where $l$ is the label number and $n$ is the words number. For the text with length $2r + 1$ and centered on $i$, the local matrix block $\overline{A_{i-r,i+r}}$ is used to measure the correlation between label phrase pairs. To improve the effectiveness of the sparse regularization, the maximum pool and ReLU function are used to expand the matrix $\overline{A}$ to generate the weight matrix $A^l$:

$$W^l = ReLU(\overline{A^{i-r,i+r}}W_1 + b_1) \tag{15}$$

$$m^l = max - pooling(W^l) \tag{16}$$

$$A^l = softmax(m^l) \tag{17}$$

Similar to the document attention mechanism, $A_l$ is used as the weight matrix to extract the document-weighted features and obtain the label-based text representation $M^l \in R^{l \times 2u}$:

$$M^l = A^l H^T \tag{18}$$

**2.1.5 Adaptive text representation fusion mechanism.**   For input LSTM hidden state $H$ and label embedding vector $C$, $M^s$ and $M^l$ are derived respectively from the document and label attention mechanisms. They are both label-related text representations. However, $M^s$ focuses on document content, and $M^l$ focuses on label-document semantic links. Use adaptive fusion mechanism to weight the two representations and obtain a specific label-based text representation $M$ containing crucial semantics for classification.

$M^s$ and $M^l$, having identical dimensions and expression forms as document feature text sequences, undergo fusion through nonlinear addition:

$$M = \lambda^s M^s + \lambda^l M^l \tag{19}$$

where $\lambda^s \in R^l$ and $\lambda^l \in R^l$ respectively represent the importance of document-based text sequence representation $M^s$ and label-based text sequence representation $M^l$ to the text sequence finally used for classification, and restrict them:

$$\lambda^s + \lambda^l = 1 \tag{20}$$

where $\lambda_i^s$ and $\lambda_i^l$ respectively represent the proportion of document-based text representation and label-based text representation in the text sequence representation of label *i*. The text sequence vector *m* extracted along the label *i* for classification can be expressed as:

$$m = \lambda_i^s M^s + \lambda_i^l M^l \tag{21}$$

The values of score proportions $\lambda^s$ and $\lambda^l$ are obtained by using the activation function $\varphi$ in combination with $M^s$ and $M^l$, where $W_{s3}$ and $W_l$ are the trained weight matrices:

$$\lambda^s = \varphi(W_{s3}M^s) \quad \lambda^l = \varphi(W_l M^l) \tag{22}$$

The text representation algorithm based on document and label information is presented in Algorithm 1. Firstly, the number of training rounds and counts per processing is determined based on the number of training samples, as described in lines 1 to 2, followed by setting the number of cycles. Word embedding is applied to the input document space *X* and label space *Y* to obtain their respective word embedding matrices, as described in lines 3 to 4. Subsequently, long-distance features are extracted from documents using a Bi-LSTM [9] network, as described in line 5. The document attention mechanism is employed to learn the document-based text representation from the content of documents, as described in lines 6 to 7. Furthermore, the label attention mechanism establishes semantic associations between labels and documents for learning the label-based text representation, as described in lines 8 to 12. Finally, an adaptive fusion mechanism allows the independent selection of proportions between these two representations to obtain a final text representation for classification prediction, as described in lines 13 to 17.

**Algorithm 1** Text Representation Algorithm based on Document and Label Information

```
Input: MLTC problem document space X and label space Y, where for
       multi-label instance {Xᵢ, Yᵢ}, Xᵢ represents document instance
       {X₁, X₂, ..., X_T}, and Yᵢ represents label set {y₁, y₂, ..., y_L}.
Output: A label-specific representation M of text with important
        semantics for classification
1: for _ in [1, ..., epoch] do
2:   for _ in [1, ..., batch] do
3:     E = Embed_word(X)
4:     C = Embed_Label(Y)
5:     H = Bi - LSTM(E)
6:     Aˢ = softmax(W_{s2}tanh(W_{s1}Hᵀ))
7:     S = AˢHᵀ
8:     A = CH
9:     Wˡ = ReLU(A_{i-r, i+r}W + b)
10:    mˡ = max - pooling(wˡ)
11:    Aˡ = softmax(mˡ)
12:    L = AˡHᵀ
13:    M₁ = sigmoid(W₁S)
```

```
14:      M₂ = sigmoid(W₂L)
15:      M₁ = M₁/(M₁ + M₂)
16:      M₂ = 1 - M₂
17:      M = M₁S + M₂L
18:   end for
19: end for
```

## 2.2 Correlation networks based on label distribution

Deep neural networks and the attention mechanism are utilized for document feature extraction. The label distribution-based correlation residual network is introduced to mitigate training costs and network degradation. It annotates text sequences with a relevant label subset and uses related knowledge to discern label correlations, enhancing the classification probability of relevant labels and reducing the irrelevant ones.

The initial classification result vector $x$ is obtained from specific label-based text representation $M$ using a fully connected layer and an output layer. Then, the classification results are enhanced through correlation to obtain the enhanced classification result vector $y$ with label distribution. $W \in R^{l \times 2u}$ is the fully connected layer's parameters, $w$ is a $l$-length vector:

$$x = wRELU(WM^T) \tag{23}$$

Each block includes a residual map with a convolution and exponential linear unit, corresponding to the left part as $f = F(x)$, and a direct map, corresponding to the right as $f = x$. The initial classification result vector $x$ and the enhanced classification result vector $y$ share the same dimensions without extra parameters or complexity. The overall function map can be expressed as:

$$y = F(x) + x \tag{24}$$

When using residual network [28], the $L + 1$ layer input can be expressed as $x_{l+1} = F(x_l, w_l) + x_l$ by the $L$ layer, and the L layer input can be expressed as $x_l = F(x_{l-1}, w_{l-1}) + x_{l-1}$ by the $L - 1$ layer. Therefore, the $L + 1$ layer input can be expressed by the $L$ and the $L - 1$ layer together, that is, $x_{l+1} = F(x_{l-1}, w_{l-1}) + F(x_l, w_l) + x_{l-1}$. By reasoning, the $L$ layer input can be expressed as $x_l = \sum_{i=1}^{L-1} F(x_i, y_i) + x_1$ by the $L-1$ layer front. Since the base map is $F(x) + x$ and the chain derivative result is:

$$\frac{\partial C}{\partial x_1} = \frac{\partial C}{\partial x_L}\frac{\partial x_L}{\partial x_1} = \frac{\partial C}{\partial x_L}\left(1 + \frac{\partial}{\partial x_1}\sum_{i=1}^{L-1} F(x_i, w_i)\right) \tag{25}$$

When a deeper network converges gradually, the presence of 1 in the equation ensures that accuracy doesn't degrade rapidly with increasing network depth. This allows for continuous parameter updates of nodes, preventing gradient vanishing or explosion. The output classification result vector $x$ in models lacking residual blocks directly trains the loss function. Residual blocks' skipped connections can address the optimization degradation issue, and the label correlations can be captured by the function $F(x)$.

Both $x$ and $y$ represent classification result vectors of the same meaning and dimension. The straightforward application of the function $F$ is a fully connected layer: $F(x) = W\sigma(x) + b$. However, this approach isn't practical for large-scale labels. Many labels exhibit no correlations, leading to correlation parameters 0 and resource wastage. One layer of residual blocks can be considered for dense labels with a limited number. In theory, increasing the number of residual blocks improves model accuracy. To balance training cost and classification accuracy, we set the function $F$ as two layers of residual blocks, with $W_1$ and $W_2$ as weight functions, $b_1$

and $b_2$ as bias terms, $\sigma$ as the sigmoid activation function, and $\delta$ as the extra exponential linear unit (ELU) function. The specific calculation formula is as follows:

$$F(x) = W_2\delta(W_1\sigma(x) + b_1) + b_2 \tag{26}$$

The correlation residual block input $x$ can be the initial classification result vector or the enhanced one. Therefore, multiple blocks can be stacked to capture intricate label correlations.

## 2.3 Optimization of classification results based on label imbalance

In large-scale label space of MLTC, label imbalance significantly affects accuracy, necessitating remediation. Consequently, the classification result optimization module addresses this via cost-sensitive strategies, employing multi-label classification loss to mitigate disparities in label and intra-label sample distributions.

**2.3.1 Multi-label classification loss function format.** From multi-classification to multi-label classification, the most direct adaptability is transformed into extending the traditional cross-entropy loss function (CE loss). For the space $D = \{(x_i, y_i) \mid 1 \leq i \leq n\}$ with $n$ samples, where $Y_i = \{y_{i1}, y_{i2}, \ldots, y_{il}, y_{ij} \in \{0, 1\}\} \subseteq Y$ is the subset of the label space and $z_i = (z_{i1}, z_{i2}, \ldots, z_{iL})$ is the classifier output. $z_{ij} \in (0, 1)$ represents the probability that sample $i$ is predicted as label $j$ and $L$ represents the label number. Using the extended Categorical Cross-Entropy (CCE) loss function directly for multi-label classification:

$$\min_\theta -\frac{1}{n}\sum_{i=1}^n\sum_{j=1}^L y_{ij}\log\left(p_{ij}\right) = -\frac{1}{n}\sum_{i=1}^n\sum_{j\in y_i^+}\frac{1}{|y_i^+|}\log\left(p_{ij}\right) \tag{27}$$

where $\theta$ is the parameter, $y_i^+$ represents the label set related to the sample $i$ and $p_{ij}$ represents the probability that the label set of sample $i$ contains label $j$, which is predicted by the model obtained through the softmax activation function:

$$p_{ij} = \frac{exp(z_{ij})}{\sum_{k=1}^L exp(z_{ik})} \tag{28}$$

The MLTC task aims to convert the $h(\circ)$ prediction into the scoring function $f : x \times y \rightarrow R$ for ranking and prioritizing relevant labels. The softmax function bifurcates training objectives into correct and incorrect label probabilities, and the sigmoid function focuses solely on the target label, which is more suitable for MLTC sorting issues. Thus, the binary cross-entropy (BCE) loss derived from the sigmoid activation function and CE loss function is obtained:

$$\min_\theta -\frac{1}{n}\sum_{i=1}^n\sum_{j=1}^L\left[y_{ij}\log\left(\sigma\left(p_{ij}\right)\right) + \left(1 - y_{ij}\right)\log\left(1 - \sigma\left(p_{ij}\right)\right)\right] \tag{29}$$

where $\sigma$ represents the sigmoid activation function $\sigma(x) = \frac{1}{1+e^{-x}}$, $f_{ij}$ represents the predicted output of deep neural network training. BCE loss, superior to CCE in MLTC as evidenced in XML-CNN [29], effectively prioritizes relevant labels. Where $P_t(x, y)$ signifies the edge distribution of label $t$ and $L$ is BCE loss:

$$error = E_{P_t(x,y)}L(f, y) \tag{30}$$

Assumed that label distribution overlaps as $P_t(x, y) = P_s(x, y)$, but label data is usually unbalanced. Large label spaces exacerbate this imbalance, with a disproportionate number of head label samples and a significant disparity between positive and negative samples within labels. Using BCE loss for classification treats head/tail labels and positive/negative samples

identically, leading to over-suppression by abundant head labels and negative samples. This results in skewed classification boundaries and poor model generalization, particularly affecting the recognition of low-frequency labels. We consider changing the *error* to associate with the head and tail labels:

$$
\begin{aligned}
error \quad &= E_{P_t(x,y)} L(f, y) \\[6pt]
&= E_{P_s(x,y)} L(f, y) \cdot \frac{P_t(x, y)}{P_s(x, y)} \\[6pt]
&= E_{P_s(x,y)} L(f, y) \cdot \frac{P_t(y) P_t(x \mid y)}{P_s(y) P_s(x \mid y)} \\[6pt]
&= E_{P_s(x,y)} L(f, y) \alpha(y) \varphi(x, y)
\end{aligned}
\tag{31}
$$

where $\alpha(y) = \frac{P_t(y)}{P_s(y)}$, $\varphi(x, y) = \frac{P_t(x|y)}{P_s(x|y)}$. Existing label balance loss functions usually focus on how to deal with $w(y)$ for sample number balance, often overlooking the variance of $\varepsilon(x, y)$ due to intra-label positive and negative sample discrepancies.

For the dominance of head label samples in imbalanced issues, the weight factor $\alpha$ is introduced to give reasonable attention parameter to rare labels, obtaining the re-weighted Binary Cross-Entropy Loss (R-BCE Loss):

$$
\min_\theta -\frac{1}{n} \sum_{i=1}^{n} \sum_{j=1}^{L} \alpha \left[ y_{ij} log \left( \sigma \left( p_{ij} \right) \right) \left( 1 - y_{ij} \right) log \left( 1 - \sigma \left( p_{ij} \right) \right) \right]
\tag{32}
$$

For the imbalance problem of positive-negative sample disparity, we introduce a modulation function $\varphi(\gamma)$ based on the R-BCE loss to down weight "easy-to-classify" negative instances, thereby deriving enhanced loss function for training multi-label text classifiers:

$$
L(x, y) = -\frac{1}{n} \sum_{i=1}^{n} \sum_{j=1}^{L} \alpha_{ij} \left[ y_{ij} \varphi \left( f_{ij}, \gamma \right) \sigma \left( f_{ij} \right) + \left( 1 - y_{ij} \right) \varphi \left( 1 - f_{ij}, \gamma \right) \sigma \left( 1 - f_{ij} \right) \right]
\tag{33}
$$

where $n$ is the sample number, $L$ is the label number, and $\sigma$ is the sigmoid function. $\alpha_{ij}$ is the weight factor used to deal with the imbalance of the sample number between labels. $\varphi$ is the modulation function, and $\gamma$ is the parameter used to deal with the imbalance of positive/negative sample numbers within labels.

The optimization module proposes label quantity-based weight factor $\alpha_{ij}$. Considering co-occurrence takes the ratio of the reciprocal of the sample's sampling probability and the reciprocal of valid sample number as the weight to protect tail classifiers from excessive suppression by dominant head samples. $\varphi$, based on prediction probability, addresses negative label dominance and maintains negative gradients for confusable labels to promote discriminative learning by focusing on classification difficulty and selective negative sample suppression.

**2.3.2 Weighting factor based on the number of labels.** This part proposes a label-based weighting factor, considering the label sample's sampling probability and valid sample number.

For sample sampling probability, the label data set is estimated using probability statistics. It is stipulated that the sample $k$ containing label $i$ is expressed as $x_i^k = 1$. A label is randomly selected during sample sampling, and a sample is selected from the label set. According to the

calculation model, the expected value of $x_i^k$ sampling frequency is:

$$P_i^c(x^k) = \frac{1}{C}\frac{1}{n_i} \tag{34}$$

However, in the MLTC problem, due to the existence of label co-occurrence, the selection of any sample $x_i^k$ does not only mean the sampling of label $i$ but means the sampling of the label set $I \subseteq Y$ related to the sample $x$. For a sample $x^k$, the sampling frequency of the sample $x^k$ is the sampling frequency of the relevant label set, that is, the sum of the sampling frequencies of all labels within the label:

$$P^I(x^k) = \sum_{y_i \in I} P_i^c(x^k) = \frac{1}{C}\sum_{x_i^k=1}\frac{1}{n_i} \tag{35}$$

Define a rebalanced weight factor $r_i^k$ to minimize the difference between the expected and actual sampling times:

$$\overline{\alpha_i^k} = \frac{P_i^c(x^k)}{P^I(x^k)} = \frac{1}{n_i}\left(\sum_{x_i^k=1}\frac{1}{n_i}\right)^{-1} \tag{36}$$

However, when sample $k$ contains both head label $i$ and tail label $j$, the ratio of the expected value $P_i^c(x^k)$ of the head label's sampling frequency and the expected value $P_j^c(x^k)$ of the tail label's sampling frequency is the reciprocal of label sample number. Due to the vast difference in the number of labels in MLTC, if the sample number of label $i$ is a hundred times that of label $j$, then $\overline{\alpha_i^k}$ will be less than 0.01. In particular, when sample $k$ contains multiple labels, it will even approach 0, making the optimization more difficult, which is not conducive to model training. To ensure the stability of the optimization process, $\overline{\alpha_i^k}$ is mapped to a reasonable range by mapping, where $\alpha$ is the overall increase of weight, $\beta$ and $\mu$ control the range of the mapping function:

$$\alpha_i^k = k + \frac{1}{1 + exp(-\beta \times (\overline{\alpha_i^k} - \mu))} \tag{37}$$

From the perspective of the number of label-valid samples, CB loss [30] associates samples with their neighborhoods to measure whether there is an overlap in data. At the same time, the valid sample number is calculated in a manner that the model and loss are unknown, and the balancing weighting factor is set as the inverse ratio of the valid sample number.

The valid number of samples is expressed as $E_n$, where $n \in Z_{>0}$ represents the number of samples. The valid number of samples is $E_n = \frac{1-\beta^n}{1-\beta}$, where $\beta = \frac{N-1}{N}$. When $\beta = 0(N = 1)$, $E_n = 1$ and when $\beta \to 1(N \to \infty)$, $E_n \to n$. The reciprocal of the label-related valid sample number is used as a weighting factor:

$$\alpha_i = \frac{1}{E_{n_i}} = \frac{1-\beta}{1-\beta^{n_i}} \tag{38}$$

where $n_i$ represents the actual sample number of label $i$, and $E_{n_i}$ represents the valid sample number of the obtained label in $i$-th class.

**2.3.3 Modulation function based on prediction probability.** This section calculates the modulation function using prediction probability from the perspective of label classification

difficulty and selective suppression of negative samples. It also performs discriminative learning by retaining negative gradients for easily confused labels.

Starting from the difficulty of distinguishing classification, the model pays more attention to the hard-to-classify samples in the label space by reducing the weight of easy-to-classify samples within the label during training. Focal loss [31] proposes a modulation function $(1 - P_t)^\gamma$ based on prediction probability, where the focus $\gamma$ is customized.

For $\gamma \in [0, 5]$, the modulation factor decreases the loss impact for well-classified samples, especially as $p_t$ approaches 1. This reduction scales with $\gamma$: at $\gamma = 0$, it equates to standard cross-entropy loss, while higher values increase the focus on hard-to-classify samples. For instance, with $\gamma = 2$, easy-to-classify samples ($p_t = 0.9$) have a loss 100 times lower than standard cross-entropy. In contrast, more challenging samples maintain a controlled loss increase, prioritizing them during training.

For negative samples, BCE loss yields a negative suppression gradient enforcing classifier $i$ to exhibit low confidence, which is beneficial to some extent. However, excessive suppression from the head label notably impedes tail label activation. This leads to tail label classifiers distancing from numerous negative samples to minimize loss, thereby overfitting to a scant count of positive samples in the feature space. The gradient of BCE loss concerning network prediction output is:

$$\frac{\partial L_{BCE}}{\partial p_{ij}} = \begin{cases} \sigma(p_{ij}) - 1, y_{ij} = 1 \\ \sigma(p_{ij}), y_{ij} = 0 \end{cases} \tag{39}$$

To enhance tail label classifier training, a selection mechanism based on prediction probability is proposed. This employs $w_{ij}$ to determine label suppression application. For confusable labels $i$ and $j$, if $p_{ij} > \xi$, suppression is activated ($w_{ij} = 1$). Otherwise, it's disregarded ($w_{ij} = 0$) to prevent excessive negative suppression:

$$w_{ij} = \begin{cases} 1, y_{ij} = 1 \\ 1, \xi \leq y_{ij} < 1 \\ 0, y_{ij} < \xi \end{cases} \tag{40}$$

The gradient of the loss relative to the network predicted output can be rewritten as:

$$\frac{\partial L}{\partial p_{ij}} = \begin{cases} \sigma(p_{ij}) - 1, y_{ij} = 1 \\ w_{ij}\sigma(p_{ij}), y_{ij} = 0 \end{cases} \tag{41}$$

The introduction of selection term $w_{ij}$ maintains the gradient for positive samples $y_{ij} = 1$ of label $i$. But for negative samples $y_{ij} = 0$ of labels $i$, label $j$ with prediction probability $p_{ij}$ above threshold $\xi$ remain unaffected. This ensures that labels potentially relevant to the current document aren't suppressed, while those with minimal correlation are disregarded, thus preventing undue negative influence.

## 3 Results

This section analyzes the experimental results on the benchmark dataset, including comparative experiments with existing models and ablation experiments of each component of LACN.

### 3.1 Dataset

This paper validates LACN on four MLTC datasets with different numbers and sizes of labels:

AAPD (https://github.com/lancopku/SGM) [32] contains 55,840 computer science academic paper abstracts, categorized into 54 topics. The AAPD is split into 54,800 training and 1,000 test samples, with a 2,000-sample validation subset. Labels are partitioned into head labels including 19 topics with a sample size greater than 2 200, middle labels including 17 topics with a sample size between 900 and 2 200, and tail labels including 18 topics with a sample size less than 900.

RCV1-v2 (https://trec.nist.gov/data/reuters/reuters.html) [33] comprises 804,414 news articles, classified into 103 topics. They are divided into 23,149 training and 781,265 test texts, with a 1,000-text validation subset. Labels are segmented into head labels including 18 topics with a sample size greater than 348, middle labels including 20 topics with a sample size between 79 and 348, and tail labels including 16 topics with a sample size less than 79.

EUR-LEX (http://manikvarma.org/downloads/XC/XMLRepository.html) [34] includes 3,956 EU legal topics, with 11,585 training and 3,865 test texts.

AmazonCat-13K (http://manikvarma.org/downloads/XC/XMLRepository.html) [35] features 1,186,239 training and 306,782 test samples from Amazon.com, categorized into 13,330 labels.

AAPD and RCV1-v2 are conventional datasets with fewer labels, while EUR-LEX and AmazonCat-13K are extreme datasets with more labels. Table 2 details the number of training samples (NtrS), the number of testing samples (NteS), the total number of words (TNW), the total number of labels (TNL), the average number of labels per sample (ANLS), the average number of samples per label (ANSL), the average number of words per training set (ANWTrS) and the average number of words per testing set (ANWTeS).

### 3.2 Testing environment

The model presented in this paper is implemented in Python and executed on Ubuntu 18.04.4 LTS. Refer to Table 3 for the detailed experimental environment:

### 3.3 Parameter configuration

For each dataset, documents are truncated after 500 words and padded zeros at the end to expand documents with less than the predefined number of words. Using the word2vec [27], document and label word embedding matrices are trained with an embedding space size of 300, fine-tuned during training. The training batch size is 64 for AAPD and EUR-LEX and 256 for RCV1-v2 and AmazonCat-13K. Training set Adam optimization learning rate to 0.001

**Table 2. Benchmark dataset properties.**

| Dataset | AAPD | RCV1-v2 | EUR-LEX | AmazonCat-13K |
|---|---|---|---|---|
| NtrS | 54 840 | 23 149 | 15 449 | 1 186 239 |
| NteS | 1 000 | 781 265 | 3 865 | 306 782 |
| TNW | 69 399 | 47 236 | 171 120 | 203 882 |
| TNL | 54 | 103 | 3 956 | 13 330 |
| ANLS | 2.41 | 3.18 | 5.32 | 5.04 |
| ANSL | 2 444.04 | 729.67 | 15.59 | 44 857 |
| ANWTrS | 163.42 | 259.47 | 1 225.20 | 246.61 |
| ANWTeS | 171.65 | 269.23 | 1 248.07 | 245.98 |

**Table 3. Laboratory test environment.**

| Hardware and Software | Configuration or Parameters |
|---|---|
| CPU | Inter(R) Core(TM) i9-10900 CPU 3.70GHz |
| GPU | GeForce RTX 3090 |
| CUDA | 11.1 |
| Python | 3.6.7 |
| Pytorch | 1.9.0 |
| mxnet | 1.3.1 |

and maximum iteration epochs to 100, monitoring whether Micro-F1 on the validation set has an increase of more than 0.5%, and the training is stopped when there is no increase for 5 consecutive epochs.

To mitigate overfitting, the drop rates of post-embedding and post-Bi-LSTM layers are set to 0.2 and 0.5, respectively. In the document-label attention mechanism, the neuron weight hyperparameters $W_{s1}$ and $W_{s2}$ are set to $d_a = 200$. For AmazonCat-13K, a 512-dimensional Bottleneck layer preceded the output layer to enhance efficiency and reduce computing load. In optimizing classification results based on label imbalance, for each loss function parameter, $\gamma = 2$ is used for FL, $\beta = 0.9$ is used for CB, the selection of the parameter $w$ is determined by optimizing based on the experimental results of the F1 value, and others are computed from existing data. During training, document and label data are input. After training, the model is tested on the validation set, and the training effect of the current model is saved. The test phase employs the best-performing model in the validation phase to perform testing on the test set.

### 3.4 Evaluation metrics

The precision and recall of class $i$ label can be expressed as:

$$Precision_i = \frac{TP_i}{TP_i + FP_i} \quad Recall_i = \frac{TP_i}{TP_i + FN_i} \tag{42}$$

Where $TP_i$ is true positive, indicating that the sample with label $i$ is also predicted to have label $i$; $FP_i$ is false positive, indicating that the sample without label $i$ is predicted to have label $i$; $FN_i$ is false negative, indicating that the sample with label $i$ but predicted to have no label $i$.

Micro-F1 and Macro-F1 are usually used to evaluate the experimental results. Micro-F1 first calculates the $Presion_{micro}$ and $Recall_{micro}$ of all labels, and then calculates the value of Micro-F1 through the F1 calculation formula, considering the overall recall and accuracy of all labels:

$$Precision_{micro} = \frac{\sum_{i=1}^{n} TP_i}{\sum_{i=1}^{n} TP_i + \sum_{i=1}^{n} FP_i} \quad Recall_{micro} = \frac{\sum_{i=1}^{n} TP_i}{\sum_{i=1}^{n} TP_i + \sum_{i=1}^{n} FN_i} \tag{43}$$

$$Micro - F1 = 2 \cdot \frac{Precision_{micro} \cdot Recall_{micro}}{Precision_{micro} + Recall_{micro}} \tag{44}$$

Macro-F1 first calculates the average $Presion_{macro}$ and $Recall_{macro}$ of each label, and then obtains the value of Macro-F1 through the F1 calculation formula to calculate the average F1

value of all labels:

$$Precision_{macro} = \frac{\sum_{i=1}^{n} Precision_i}{n} \quad Recall_{macro} = \frac{\sum_{i=1}^{n} Recall_i}{n} \tag{45}$$

$$Macro - F1 = 2 \cdot \frac{Precision_{macro} \cdot Recall_{macro}}{Precision_{macro} + Recall_{macro}} \tag{46}$$

It can be seen from the above equation that Micro calculates the average value by combining the contribution of different labels and gives more weight to frequent labels, while Macro calculates each label independently and gives all labels the same weight. Therefore, Micro pays more attention to labels with fewer samples than Macro. For the MLTC problem with imbalanced labels, the Micro-F1 value is selected as the evaluation index in this paper.

In addition to measuring the accuracy and recall of labels, in particular, for large-scale labeled datasets, considering the sparsity of labels, a short list of potentially relevant labels for each test sample is used to represent the classification quality. Two sample-based ranking criteria *precision@k* and *nDCG@k* are used to evaluate the model, where $r_k \tilde{z}$ is the correct prediction value of the top *K* labels, and $\|z\|$ is the number of correctly predicted labels, with k = 3 and 5 being the most representative.

*precision@k* represents the accuracy at the *k*-th ranked label:

$$precision@k = \frac{1}{k} \sum_{l \in r_k \tilde{z}} z_l \tag{47}$$

*nDCG@k* represents the average normalized discounted cumulative return at the *k*-th ranked label:

$$DCG@k = \sum_{l \in r_k \tilde{z}} \frac{z_l}{log(l+1)} \quad nDCG@k = \frac{DCG@k}{\sum_{l=1}^{min(k,\|z\|)_0)} \frac{1}{log(l+1)}} \tag{48}$$

### 3.5 Analysis of model comparison

This paper uses various methods as a baseline, as shown below:

- SGM [32]: Utilizes a sequence generation model for the MLTC task, predicting labels while considering label correlation and vital feature selection.

- Seq2Set [16]: Employs reinforcement learning with rewards to handle label sequences' high-order correlations and commutative invariance.

- LEAM [17]: Approaches text classification via label-word joint embedding and attention-based compatibility measurement.

- ML Reasoner [36]: Implements binary classification with iterative reasoning to manage label correlations without order sensitivity.

- XML-CNN [29]: Adapts deep learning with new CNNs and dynamic pooling for advanced feature extraction in extreme multi-label text classification.

- Attention-XML [13]: Uses multi-label attention and a probability label tree to handle text relevance and scalability in a large label space.

- BBN [21]: Introduces a bilateral branching network for integrated representation and classifier learning in long-tail tasks.

**Table 4. Comparing the Precision, Recall and F1 values of the model on AAPD and RCV-v2.**

| Model | AAPD | | | RCV1-v2 | | |
|---|---|---|---|---|---|---|
| | Precision | Recall | F1 | Precision | Recall | F1 |
| SGM | 74.6 | 65.9 | 69.9 | 88.7 | 85.0 | 86.9 |
| Seq2Set | 73.9 | 67.4 | 70.5 | 90.0 | **85.8** | **87.9** |
| LEAM | 76.5 | 59.6 | 67.0 | 87.1 | 84.1 | 85.6 |
| ML-R | 72.6 | **71.8** | 72.2 | 89.0 | 85.2 | 87.1 |
| LACN | **79.7** | 67.2 | **72.9** | **91.4** | 84.6 | **87.9** |

- HTTN [37]: Enhances tail label recognition by transferring meta-knowledge from data-rich to data-poor labels.

Table 4 compares the Precision, Recall, and F1 values of SGM, Seq2Set, LEAM, ML-R, and the LACN model proposed in this paper on AAPD and RCV1-v2 datasets. The best results are highlighted in bold. The LEAM model, focusing solely on document content, underperforms due to its disregard for label correlations. In contrast, both SGM and Seq2Set are sequence generation models based on Seq2Seq and exhibit enhanced label correlation capturing. Seq2-Set's reward feedback mechanism further mitigates label order sensitivity for superior results. The ML-R model, innovatively addressing multi-label text classification through inferential label correlation, achieves higher recall. The LACN model proposed in this paper considers document content, label correlation, and label imbalance, surpassing others in Micro-F1.

Table 5 contrasts the P@k and NDCG@k values of XML-CNN, Attention-XML, BBN, HTTN, and the LACN model proposed in this paper on AAPD, RCV1-v2, EUR-LEX and AmazonCat-13K dataset. The best results are highlighted in bold. The XML-CNN excels in

**Table 5. Comparing the model's P@k and NDCG@k on AAPD, PCV1-v2 and EUR-LEX, AmazonCat-13K.**

| Dataset | Model | P@1 = N@1 | P@3 | P@5 | NDCG@3 | NDCG@5 |
|---|---|---|---|---|---|---|
| AAPD | XML-CNN | 74.38 | 53.84 | 37.79 | 71.12 | 75.93 |
| | Attention-XML | 83.02 | 58.72 | 40.56 | 78.01 | 82.31 |
| | BBN | 81.56 | 57.81 | 39.10 | 76.92 | 80.06 |
| | HTTN | 83.84 | 59.92 | 40.79 | 79.27 | 82.67 |
| | LACN | **84.72** | **61.11** | **42.62** | **80.54** | **84.04** |
| RCV1-v2 | XML-CNN | 95.75 | 78.63 | 54.94 | 89.89 | 90.77 |
| | Attention-XML | 96.41 | 80.91 | 56.38 | 91.88 | 92.70 |
| | BBN | 94.61 | 77.98 | 54.25 | 88.97 | 89.68 |
| | HTTN | 95.86 | 78.92 | 55.27 | 89.61 | 90.86 |
| | LACN | **96.53** | **81.79** | **56.98** | **92.65** | **93.26** |
| EUR-LEX | XML-CNN | 76.81 | 62.79 | 51.56 | 76.81 | 60.47 |
| | Attention-XML | **85.43** | **73.30** | **60.99** | 76.54 | **70.45** |
| | BNN | 76.22 | 60.40 | 49.45 | 64.26 | 58.54 |
| | HTTN | 81.14 | 67.62 | 56.38 | 70.89 | 64.42 |
| | LACN | 82.52 | 67.48 | 58.29 | **76.93** | 64.62 |
| AmazonCat-13K | XML-CNN | 94.53 | 79.12 | 63.38 | 88.19 | 85.61 |
| | Attention-XML | 95.13 | 81.12 | 66.10 | 89.90 | 88.13 |
| | BBN | 92.98 | 79.23 | 64.62 | 87.62 | 85.93 |
| | HTTN | 93.40 | 79.83 | 64.89 | 88.01 | 87.74 |
| | LACN | **95.69** | **82.41** | **67.32** | **90.13** | **88.73** |

large-scale data handling but overlooks label correlation. Attention-XML, integrating Bi-LSTM and attention mechanisms, excels in text representation and correlation mining and particularly benefits from predefined label hierarchies in datasets like EUR-LEX. Conversely, in extensive label datasets like AmazonCat-13K, its performance is overshadowed by models with ample training samples per label. BBN and HTTN, focusing on long-tail distribution in MLTC, show better performance with larger k values but neglect label text and inter-label correlations. In contrast, LACN uses the label attention mechanism to obtain the text representation of a specific label. Even in the extreme label dataset EUR-LEX, the number of samples corresponding to the label is small and there is no sufficient number of relevant samples to support training, the existing samples can be mined for relevant semantics to obtain a representation that is more consistent with text classification. It also uses the distribution-based correlation network to introduce label correlation and optimizes for label imbalance. Even in the large-scale label dataset AmazonCat-13K, where there are many label categories and complex correlations among labels, it can get better classification results.

## 3.6 Analysis of ablation experiment comparison

This section performs ablation experiments to validate the effect and optimizations of the text representation, label distribution, and classification results in the proposed LACN model.

**3.6.1 Comparison of text representation.** This paper employs the Bi-LSTM network, fully connected layer, and BCE loss as the foundational model to compare the partial role of document and label information-based text representations. Based on the model, document attention mechanism (W), label attention mechanism (B), the average of text-based and label-based text representations summed directly (W+B), and the text representation obtained by fusing the text-based and label-based text representations through the adaptive fusion mechanism (R+W+B) are used and tested on AAPD, RCV1-v2, EUR-LEX, and AmazonCat-13K datasets. Then, the experimental classification results of P@1, P@3, and P@5 are evaluated.

Fig 3 details the performance of W, B, W+B, R+W+B, and the LACNmodel proposed in this paper by the value of P@1, P@3, and P@5 on AAPD, RCV1-v2, EUR-LEX, and Amazon-Cat-13K datasets. In the feature representation of the text content, W tends to remove the redundant text information and find the important content related to the label but does not consider the association between the document and the label text content. B focuses on the semantic relationship between the document and the label content by learning the label text to mine the document semantics most related to the label explicitly. Still, there are differences between tags that cannot be distinguished by the tag text. Through the organic combination of the two representations, the most relevant discriminative information to the corresponding label can be extracted from each document, and the adaptive fusion mechanism enables the model to adaptively select the text representation that is most beneficial to the final prediction result during the training process. Classification results show that the number of label samples in the datasets AAPD, RCV1-v2 and AmazonCat-13K is relatively dense, and each label has a sufficient number of samples for training so that more content can be learned from the text content. In contrast, in the extreme label dataset EUR-LEX, the number of labels is large and the number of samples is small. For tail labels, there is no sufficient number of relevant samples to support training. It can be of great help to mine the relevant semantics of existing samples from the perspective of text content.

**3.6.2 Comparison of label co-occurrence.** To compare the role of correlation networks based on label distributions, on the base model, single text representation (A), single relevance network (C), the combination of text representation and relevance network (A+C), and the

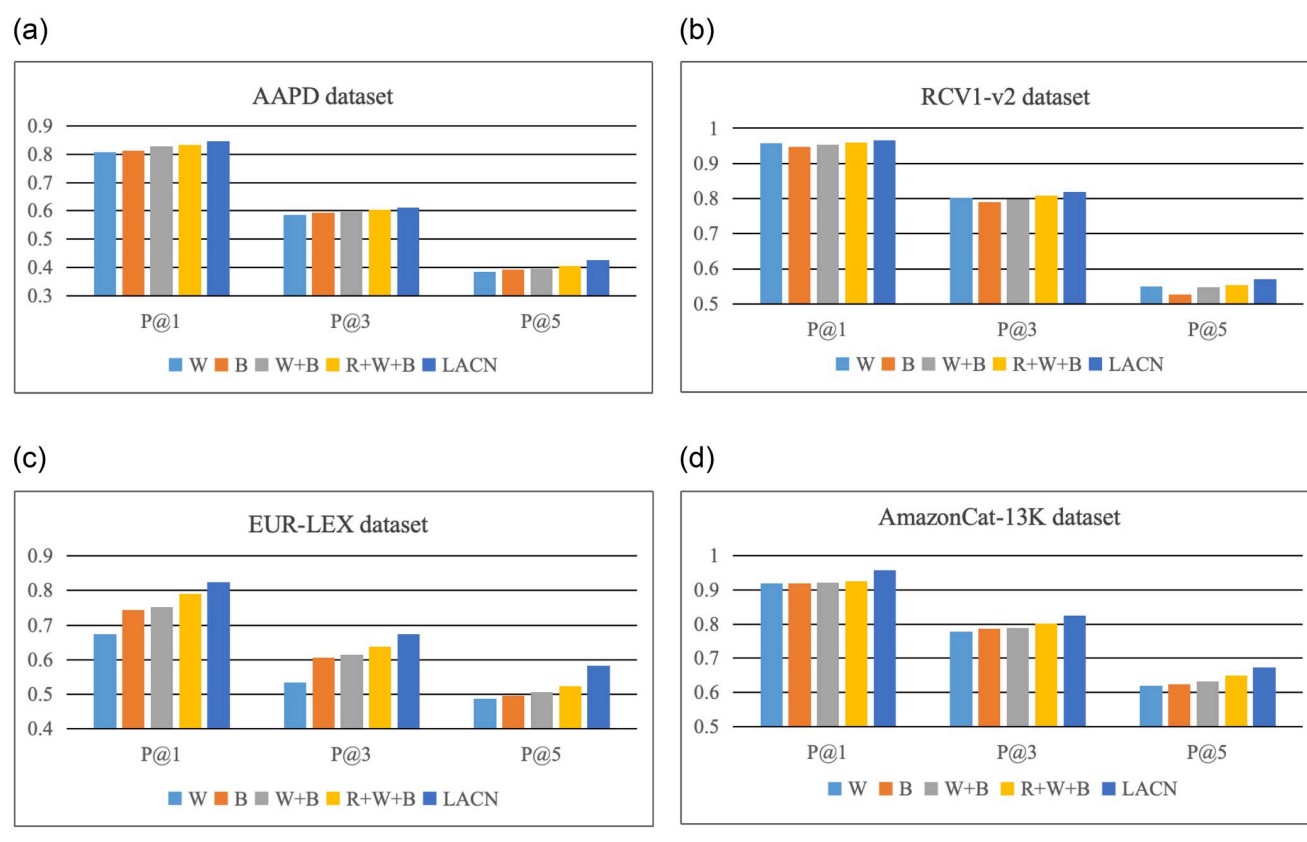

**Fig 3. Comparison of ablation experiments on text representation.**

LACN model proposed in this paper are used and trained on the AAPD, RCV1-v2, EUR-LEX, AmazonCat-13K datasets.

The experimental classification results of P@1, P@3, and P@5 are compared as shown in Fig 4. The result indicates that A neglects label associations, leading to performance degradation due to the superposition of the network. C enhances original classification results using relevant knowledge, but the complex information in documents and labels is not fully utilized. A+C effectively extracts document semantics related to the label, alleviates network degradation, and introduces label distribution, significantly improving model performance. From the perspective of classification results, the performance of the model can be significantly improved through the study of documents and label contents due to the sufficient number of samples, while the introduction of label correlation only through the use of the correlation network is slightly inadequate. However, the performance of the model can be further improved through correlation network mapping to enhance the original label prediction after text representation. In large-scale label datasets EUR-LEX and AmazonCat-13K, there are complex correlations between labels due to a large number of label categories, therefore the use of the correlation network for label correlation mining can achieve great improvement. In particular, in the EUR-LEX dataset, with numerous label categories and a few samples, using A lacks sufficient document content for effective learning, resulting in poor performance. In contrast, using C can extract information from the labels and achieve better results.

Considering training cost and classification accuracy, this paper uses two residual blocks in the correlation networks. Theoretically, more blocks enhance accuracy and escalate training

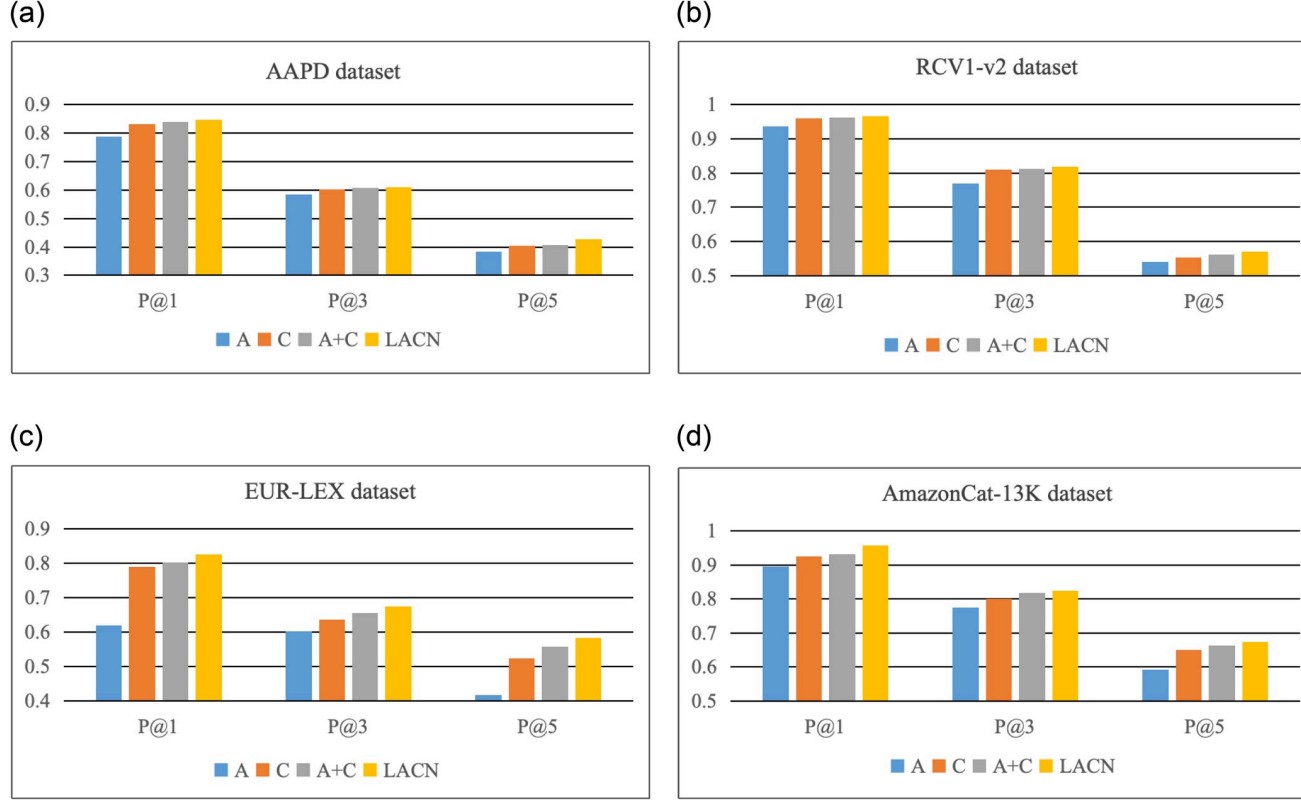

**Fig 4. Comparison of label co-occurrence ablation experiments.**

expenses with uncertain benefits. However, with the increase in the number, the training cost increases and the effect is not necessarily better. Fig 5 demonstrates that by comparing the experimental classification results of P@1, P@3, P@5, and F1 values using different numbers of residual blocks for training on AAPD and EUR-LEX datasets. In the AAPD dataset, the optimal results occur with one block, deteriorating as blocks increase. Conversely, the EUR-LEX dataset shows improved performance with more blocks. Text representation-based label feature extraction benefits text classification training for dense-label datasets like AAPD, but excessive distribution hampers model performance. Conversely, sparse-label datasets like EUR-LEX lack sufficient samples per label for effective training, therefore, more information can be obtained from the label distribution.

**3.6.3 Comparison of classification results.** The label distribution of the four datasets analyzed in this paper is unbalanced with the long-tail phenomenon, that is, a small number of labels occupy a large number of documents, while most labels have only a small number of documents associated with them. On the whole, the long-tail phenomenon of the RCV1-v2 data set is more obvious, with most labels containing less than 1/3 of the maximum sample number, and most samples appear in the top 2/5 label category groups. Obviously, the training of the last 3/5 label category group will be more difficult than the other categories due to the lack of training samples. The same long-tail distribution phenomenon also exists on the EUR-LEX and AmazonCat-13K datasets. This section mainly analyzes numerical imbalance on AAPD, RCV1-v2, and EUR-LEX, AmazonCat-13K datasets.

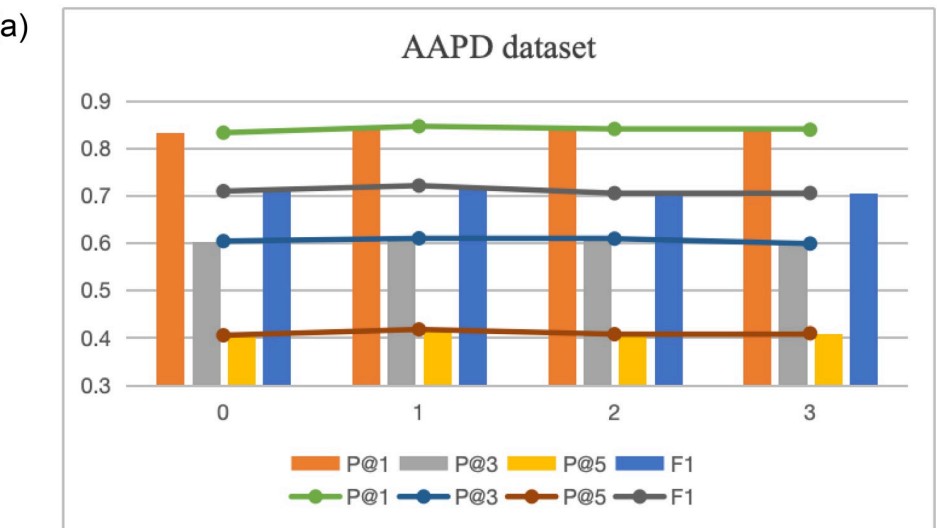

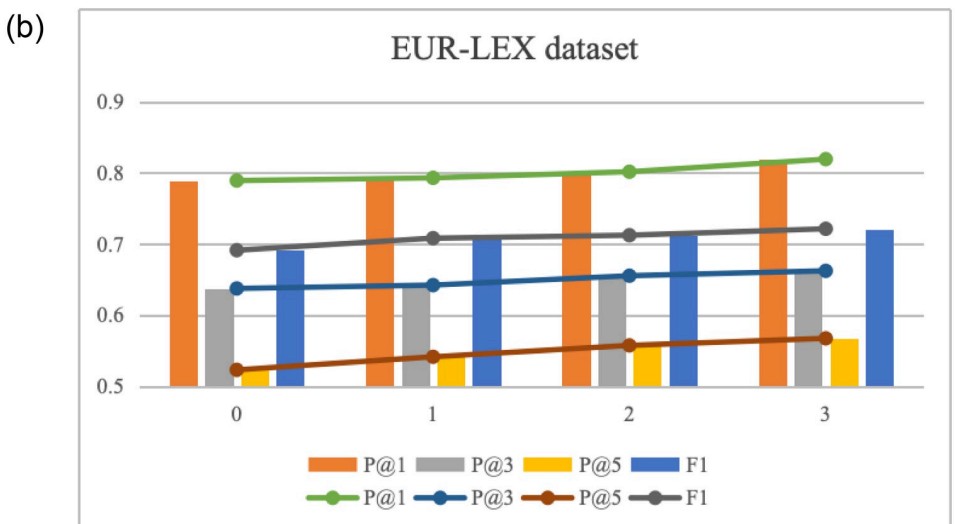

**Fig 5. P@1, P@3, P@5, and F1 values of different residual block numbers in AAPD and EUR-LEX datasets.**

The assessment of label imbalance is achieved through the analysis of $IRLbl(\lambda)$ and $MeanIR$, where $IRLbl(\lambda)$ denotes the ratio of the most prevalent label to the number of corresponding samples, indicating label sparsity. A higher $IRLbl(\lambda)$ value signifies rarer label occurrence and a more significant discrepancy with the most frequently occurring label. The function $h(\lambda, Y_i)$ determines whether the label is in the label space, with the $IRLbl(\lambda)$ minimum being 1:

$$IRLbl(\lambda) = \frac{\max_{\lambda \in L}(\sum_{i=1}^{n} h(\lambda, Y_i))}{\sum_{i=1}^{n} h(\lambda, Y_i)} \tag{49}$$

The average value $MeanIR$ of all the label imbalance rates in the label space is used to measure the average imbalance rate of the whole label set. The larger $MeanIR$ is, the greater the difference of occurrence counts between the labels in the dataset and the higher the degree of

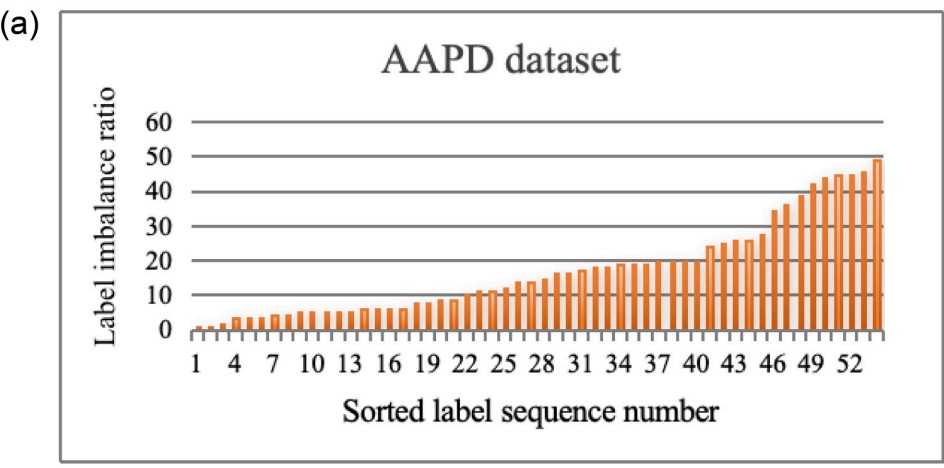

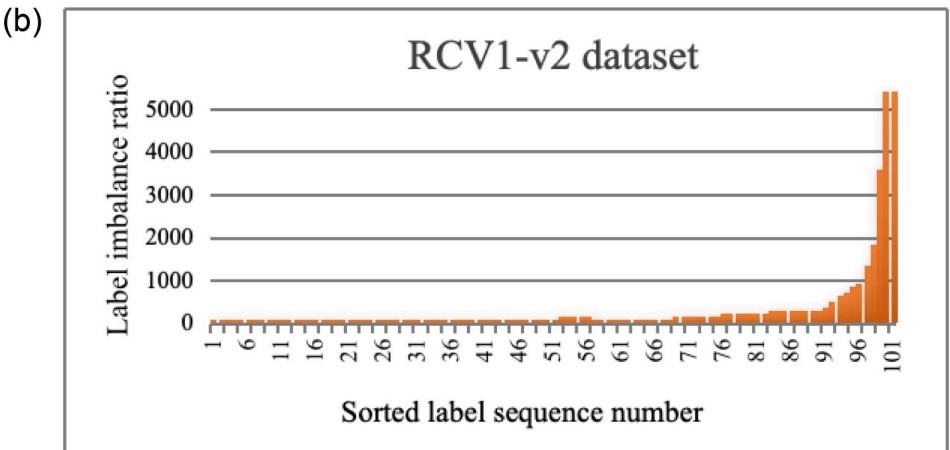

**Fig 6. Label imbalance ratio chart of AAPD and RCV1-v2 datasets.**

imbalance between the labels is. Where $q$ is the label space size:

$$MeanIR = \frac{1}{q}\sum_{\lambda \in L} IRLbl(\lambda) \tag{50}$$

Fig 6 is the label imbalance ratio graph of the AAPD and RCV1-v2 datasets. The label imbalance ratio graph sorts the label categories by the number of label-related samples and calculates the label imbalance ratio of each category respectively. As can be seen from the graph, the label with more label-related samples has a lower label imbalance ratio. In particular, as the number of labels increases, the corresponding number of tail labels increases, while the label imbalance ratio of tail labels is very low or even tends to 0, resulting in a decrease in the average imbalance ratio of the overall dataset. The same phenomenon also exists on the EUR-LEX and AmazonCat-13K datasets, but the number of label categories is too large to be displayed as a bar chart.

Table 6 numerically presents average imbalance ratios for datasets AAPD, RCV1-v2, EUR-LEX, and AmazonCat-13K. The AAPD dataset is relatively balanced. In contrast, the EUR-LEX and RCV1-v2 datasets are imbalanced at larger and smaller label spaces, respectively. The

**Table 6. Average imbalance ratios for AAPD, RCV1-v2, and EUR-LEX, AmazonCat-13K datasets.**

| Dataset | AAPD | RCV1-v2 | EUR-LEX | AmazonCat-13K |
|---|---|---|---|---|
| Average imbalance ratio | 17.041 | 274.222 | 82.846 | 49 821.016 |

AmzonCat-13K dataset is imbalanced, with huge label space. Larger label spaces tend to increase imbalance ratios due to limited sample size.

Label imbalance in the four datasets is addressed using multi-label classification loss functions. To compare the role of each imbalance parameter, four separate parameters are used based on the base model, including a label number-based weight factor computed from the perspective of the sampling probability (R) and the label's valid sample number (C), a modulation function based on the prediction probability computed from the perspective of the label-classified difficulty (F) and the negative sample's selective suppression (Y), and a loss functions (R+F, R+Y, C+F, C+Y) combined of the weighting factor and the modulation function and trained in the format of Eq 33. They are trained on AAPD, RCV1-v2 and EUR-LEX, Amazon-Cat-13K datasets. Performance is assessed by comparing the experimental classification results of P@1, P@3, P@5, and F1 values and epochs. Where R is obtained by calculating the number of samples related to the label, C uses $\gamma = 2$, F uses $\beta = 0.9$, and Y gets the best by setting different experiment values.

Fig 7 shows the experimental results of AAPD and RCV1-v2 at P@1, P@3, P@5, F1 value and numbers of the epoch. The results demonstrate that loss functions targeting label

(a)
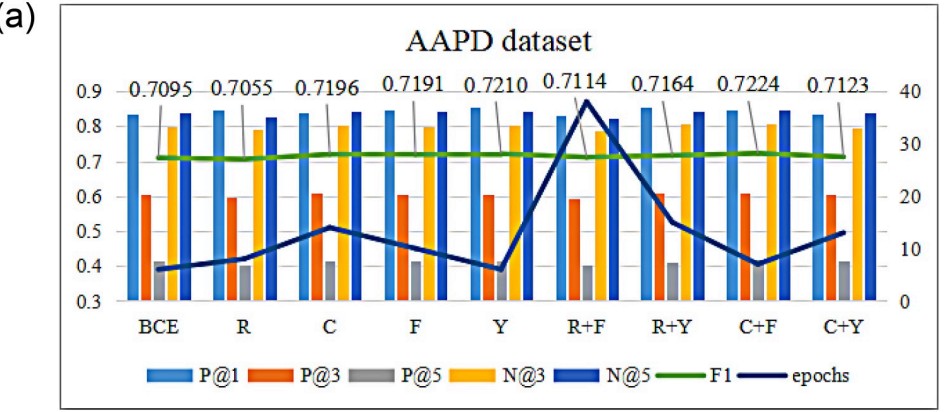

(b)
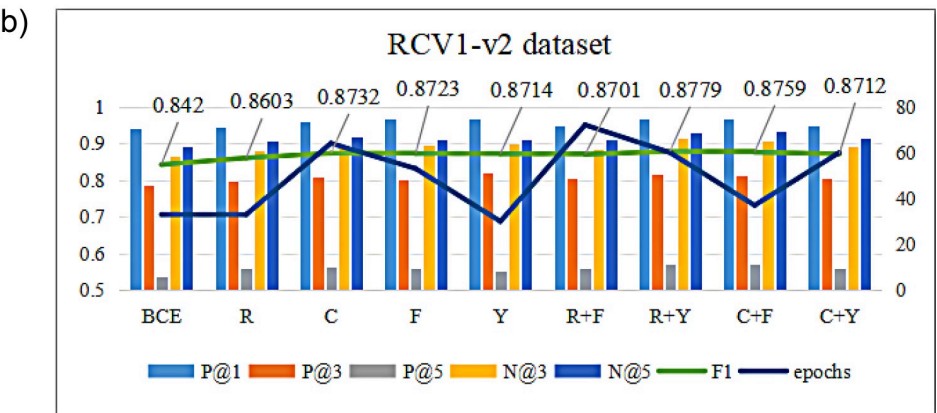

**Fig 7. Comparison of label-predicted ablation experiments.**

(a)

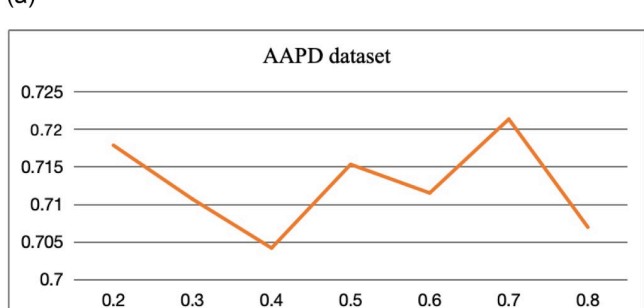

(b)

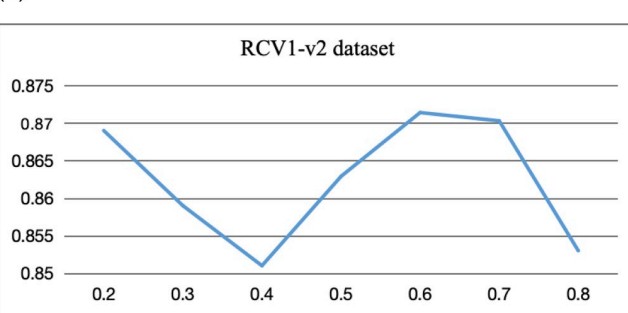

**Fig 8. F1 value variation curve when the selection items are different in AAPD and RCV1-v2 datasets.**

imbalance can improve model performance despite increasing training time. In the AAPD dataset, due to the dense data and relatively low imbalance degree, R simply based on sample sampling probability cannot optimize the model well. In addition, other loss functions yield varying enhancements. Experimental results show that the optimal combination of loss functions for AAPD is C+F, balancing high F1 and training time. In the RCV1-v2 dataset, various improved loss functions can optimize the model performance to a certain extent due to the high label imbalance. For RCV1-v2, R+Y is optimal without training time considered, and C +F is optimal with both training time and classification optimization considered. The optimal loss function may be different for different data sets, which are all determined by experiments. However, from the perspective of experimental time consumption, R+F is less recommended for being too time-consuming. Y excels with negative sample suppression. Therefore, in summary, using C, F, and Y alone or R+Y, C+F can achieve better classification optimization results in terms of experimental results and time loss.

Fig 8 illustrates the F1 value curve for different values of the selection parameter $w$ on the AAPD and RCV1-v2 datasets. It is observed that for $w \in \{0.2, 0.3, 0.4\}$, the model minimally suppresses negative samples, resulting in a lack of selective inhibition, information loss, and decreased model performance. As $w$ transitions to $\{0.5, 0.6, 0.7\}$, it effectively selects inhibitory labels, enhancing the training of the tail label classifier. However, for $w = 0.8$, the model disregards all negative samples, leading to overfitting on positive samples and a significant decline in training efficacy. Consequently, we set the selection parameter for model training to $w \in \{0.5, 0.6, 0.7\}$, with the optimal value determined as the outcome.

## 4 Limitations

In practical applications, because label attention and correlation networks usually need to compute and update a large number of labels, the training time of the model is long, which limits the efficiency of the model. The assumption of the correlation between labels is simple. However, in the real world, the correlation between labels may be more complex, and the correlation network based solely on the distribution of labels may not capture the complex relationship between labels well.

## 5 Conclusions and future work

This paper presents a new framework LACN to solve the multi-Label text classification problem. The main contributions are as follows: Firstly, the LACN utilizes the label attention mechanism to obtain the most relevant classification information for the texts. In addition, a

correlation residual network is introduced to learn the label co-occurrence and improve the accuracy of label prediction. Finally, we propose using a weighting factor and a modulation function to adjust the cost-sensitive loss function thereby tackling the problem of the imbalanced number of samples for each label in the MLTC.

The experiment uses four multi-label text datasets. The comparative experimental results show that LACN can achieve optimal or suboptimal results versus state-of-the-art methods. On the AAPD data set, compared with the suboptimal method, it outperforms the second-best method by 2.05% $\sim$ 5.07% in precision@k and by 2.10% $\sim$ 3.24% in NDCG@k for k = 1, 3, 5. The ablation experiment verifies that all components of LACN are essential for its success.

Future work can be carried out from the following three aspects. Firstly, pre-trained language models like BERT can be used to process text to enhance text representation capabilities. Secondly, the correlation network based on label distribution can be extended by incorporating a graph neural network so that it can not only make use of label distribution but also handle increasingly complex structural relationships in the real world, such as inclusion, overlap, etc. This will enable handling complex structural relationships in the real world. Thirdly, developing flexible loss functions can help to adapt to different training sets without requiring additional parameter statistics.

## Author Contributions

**Conceptualization:** Ling Yuan, Ping Sun.

**Data curation:** Ling Yuan.

**Formal analysis:** Ling Yuan.

**Funding acquisition:** Ping Sun.

**Investigation:** Hai ping Yu, Yin Zhen Wei.

**Methodology:** Ling Yuan, Hai ping Yu.

**Project administration:** Ping Sun.

**Software:** Xinyi Xu.

**Visualization:** Jun jie Zhou.

**Writing – original draft:** Xinyi Xu.

**Writing – review & editing:** Xinyi Xu.

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
