## [Decision Letter · Decision Letter 0]

1 Aug 2024

PONE-D-24-27919Research of Multi-Label Text Classification based on Label Attention and Correlation NetworksPLOS ONE

Dear Dr. Sun,

Thank you for submitting your manuscript to PLOS ONE. After careful consideration, we feel that it has merit but does not fully meet PLOS ONE’s publication criteria as it currently stands. Therefore, we invite you to submit a revised version of the manuscript that addresses the points raised during the review process.

We look forward to receiving your revised manuscript.

Kind regards,

Tianlin Zhang

Academic Editor

PLOS ONE

Journal Requirements:

   "This paper is funded by the National Natural Science Foundation of China under

Grant No.62272180, Hubei Provincial Teaching and Research Project for Higher

Education Institutions (No.2022570), Wuhan Education Science Planning Project

(No.2022C151), Wuhan Vocational College of Software and Engineering Research

Startup funding project (Grant No.KYQDJF2023004), Wuhan Vocational College of Software and Engineering 2023 Doctor Team Science and Technology Innovation Platform Project (No.BSPT2023001)."

6. Please provide a complete Data Availability Statement in the submission form, ensuring you include all necessary access information or a reason for why you are unable to make your data freely accessible. If your research concerns only data provided within your submission, please write "All data are in the manuscript and/or supporting information files" as your Data Availability Statement.

7. When completing the data availability statement of the submission form, you indicated that you will make your data available on acceptance. We strongly recommend all authors decide on a data sharing plan before acceptance, as the process can be lengthy and hold up publication timelines. Please note that, though access restrictions are acceptable now, your entire data will need to be made freely accessible if your manuscript is accepted for publication. This policy applies to all data except where public deposition would breach compliance with the protocol approved by your research ethics board. If you are unable to adhere to our open data policy, please kindly revise your statement to explain your reasoning and we will seek the editor's input on an exemption. Please be assured that, once you have provided your new statement, the assessment of your exemption will not hold up the peer review process.

Reviewers' comments:

Reviewer's Responses to Questions

**Comments to the Author**

1. Is the manuscript technically sound, and do the data support the conclusions?

Reviewer #1: Yes

Reviewer #2: Yes

2. Has the statistical analysis been performed appropriately and rigorously? 

Reviewer #1: Yes

Reviewer #2: Yes

3. Have the authors made all data underlying the findings in their manuscript fully available?

Reviewer #1: Yes

Reviewer #2: Yes

4. Is the manuscript presented in an intelligible fashion and written in standard English?

Reviewer #1: Yes

Reviewer #2: Yes

5. Review Comments to the Author

Reviewer #1: This manuscript introduces a label attention and correlation networks model to address the challenges of classifying multi-label text and enhance classification performance. Adequate revisions to the following points should be undertaken to justify the recommendation for publication.

Proofread the manuscript carefully to eliminate any grammatical errors or typos and ensure clarity and coherence in writing. Additionally, adhere to the formatting and style guidelines specified by the target journal or publication venue to enhance the professionalism of the manuscript.

The abstract section is fragile. Please re-write an abstract section, explain the result obtained and contribution, improve a proposed method, etc. Please delete unnecessary information.

The authors should clearly state the limitations of the proposed method in other real applications.

I suggest the authors add a table at the end of the literature review and compare the reviewed papers to clarify the research gap better.

Please write your contribution to this paper in the Introduction section.

Please draw a flowchart of the proposed method.

How did the authors set parameters for their proposed algorithm? Please make sensitivities of these parameters to the performance of their proposed algorithm!

Incorporating relevant and recent academic sources could strengthen your paper's validity and give readers more context and background. a hybrid model based on convolutional neural network and long short-term memory for multi-label text classification, an automatic text summarization based on valuable sentences selection,

Expand the critical results in the conclusion. Focus on the main developments in the finale. Also, write the main contributions in the conclusion.

Numerical results are good enough, but more explanations are required to analyze each figure presented.

Please Change the “conclusion” section to “ Conclusion and Future Work” and write future work.

All figures are of low quality, so please improve all of them.

Good luck

Reviewer #2: I have gone through the manuscript for clarity and have some suggestions for improvement as follows:

1. Please include the full equation of the evaluation metrics.

2. Have you considered the overlapping between labels?

3. Was the proposed method suitable for addressing the problem on all datasets? If so, why? Were all datasets imbalanced? If so, which one was the hardest to balance?

4. Please add critical analysis at the end of the previous studies to demonstrate the main motivation of the current study.

6. PLOS authors have the option to publish the peer review history of their article (what does this mean?). If published, this will include your full peer review and any attached files.

Reviewer #1: No

Reviewer #2: No

---

## [Author Response · Author response to Decision Letter 0]

7 Sep 2024

Rseply to Reviewer 1:

1-Proofread the manuscript carefully to eliminate any grammatical errors or typos and ensure clarity and coherence in writing. Additionally, adhere to the formatting and style guidelines specified by the target journal or publication venue to enhance the professionalism of the manuscript.

reply: Corrections in various sections. For example: downweight → down weight, To summary → In summary. Modify the structure style according to the target journal. For example, change the title of section 2 from Multi-Label Text Classification based on Label Attention and Correlation Networks to Method, and change the title of section 3 from Experimental Section to Results.

2-The abstract section is fragile. Please re-write an abstract section, explain the result obtained and contribution, improve a proposed method, etc. Please delete unnecessary information.

reply: Rewrite the abstract section.

3-The authors should clearly state the limitations of the proposed method in other real applications.

Reply: Add the limitations of the proposed method at Section 4. For example, the correlation network based solely on the label distribution may not capture the complex relationship between labels well.

4-I suggest the authors add a table at the end of the literature review and compare the reviewed papers to clarify the research gap better.

Reply: Add a table to compare the reviewed papers at paragraph 5 of Section 1 Introduction.

5-Please write your contribution to this paper in the Introduction section.

Reply: Write the main three contributions to this paper at the penultimate paragraph of the Introduction section.

6-Please draw a flowchart of the proposed method.

Reply: Add a flowchart of the proposed method at paragraph 1 of Section 2 Methods.

7-How did the authors set parameters for their proposed algorithm? Please make sensitivities of these parameters to the performance of their proposed algorithm!

Reply: As described in Section 3.3 Parameter Configuration, authors first selected a set of default parameters and then adjusted these parameters by experimenting with different parameter values to get the best performance. Add the analysis of the impact of varying the selection parameter on model performance.

8-Incorporating relevant and recent academic sources could strengthen your paper's validity and give readers more context and background. a hybrid model based on convolutional neural network and long short-term memory for multi-label text classification, an automatic text summarization based on valuable sentences selection.

Reply: Add literatures [6] an automatic text summarization based on valuable sentences selection and [12] a hybrid model based on convolutional neural network and long short-term memory for multi-label text classification at paragraph 1-2 of Section 1 Introduction.

9-Expand the critical results in the conclusion. Focus on the main developments in the finale. Also, write the main contributions in the conclusion.

Reply: Add the critical results at paragraph 2 of Section 4. Simplify the content to focus on the main developments of the proposed method and write the main contributions of the proposed method in the paper at paragraph 1 of Section 4.

10-Numerical results are good enough, but more explanations are required to analyze each figure presented.

Reply: Add a more detailed description of the experimental data results shown in Figure 2, Figure 4 and Figure 6.

11-Please Change the “conclusion” section to “ Conclusion and Future Work” and write future work.

Reply: Change the title of section 4 from Conclusions to Conclusion and Future Work and write the future work from what aspects to do.

12-All figures are of low quality, so please improve all of them.

Reply: All figures were improved by converting to vectors or adjusting the resolution.

Reply to Reviewer 2:

1-Please include the full equation of the evaluation metrics.

Reply: Add the full equation of the evaluation metrics at Section 3.4.

2-Have you considered the overlapping between labels?

Reply: We have not yet considered the overlap between labels in the current study, but we will consider it as a future work to extend the correlation network based on label distribution, for example, using the graph neural network to learn the internal relations of labels and handle increasingly complex structural relationships between labels in the real world, such as inclusion and overlap.

3-Was the proposed method suitable for addressing the problem on all datasets? If so, why? Were all datasets imbalanced? If so, which one was the hardest to balance?

Reply: Yes, the proposed method was suitable for addressing the problem on all datasets because the method considers document content, label correlation, and label imbalance comprehensively. It uses the label attention mechanism to obtain the text representation of a specific label, uses the distribution-based relevance network to introduce label correlation, and optimizes for label imbalance, which can get the most and sub-optimal results on Micro-F1, P@k and NDCG@k compared with other methods. The detailed presentation and experimental data are shown in Table 3 and Table 4 of paragraph 2-3 of Section 3.5 on page 18-19. The RCV1-v2 dataset was the hardest to balance due to the long-tail phenomenon as described in paragraph 1 of Section 3.6.3 on page 22.

4-Please add critical analysis at the end of the previous studies to demonstrate the main motivation of the current study.

Reply: Add critical analysis of the previous studies from aspects of text representation, label correlation and data imbalance in paragraph 5 of Section 1 on page 2.

---

## [Editor Report · Decision Letter 1]

13 Sep 2024

Research of Multi-Label Text Classification based on Label Attention and Correlation Networks

PONE-D-24-27919R1

Dear Dr. Sun,

We’re pleased to inform you that your manuscript has been judged scientifically suitable for publication and will be formally accepted for publication once it meets all outstanding technical requirements.

Kind regards,

Tianlin Zhang

Academic Editor

PLOS ONE
---

## [Editor Report · Acceptance letter]

18 Sep 2024

PONE-D-24-27919R1 

PLOS ONE

Dear Dr. Sun, 

I'm pleased to inform you that your manuscript has been deemed suitable for publication in PLOS ONE. Congratulations! Your manuscript is now being handed over to our production team.

Kind regards, 

on behalf of

Dr. Tianlin Zhang 

Academic Editor

PLOS ONE